# Obesity phenotypes and dyslipidemia in adults from four African countries: An H3Africa AWI-Gen study

Engelbert A. Nonterah[1,2,3]*, Godfred Agongo[1,4], Nigel J. Crowther[5], Shukri F. Mohamed[6], Lisa K. Micklesfield[7], Palwendé Romuald Boua[8,9], Alisha N. Wade[10], Solomon S. R. Choma[11], Hermann Sorgho[8], Isaac Kissiangani[6], Gershim Asiki[6], Patrick Ansah[1], Abraham R. Oduro[1], Shane A. Norris[7], Stephen M. Tollman[10], Frederick J. Raal[12], Marianne Alberts[11†], Michele Ramsay[9]*, as members of AWI-Gen and the H3Africa Consortium

1 Navrongo Health Research Centre, Ghana Health Service, Navrongo, Ghana, 2 Julius Centre for Health Sciences and Primary Care, University Medical Centre Utrecht, Utrecht University, Utrecht, The Netherlands, 3 Department of Epidemiology, School of Public Health, C.K. Tedam University of Technology and Applied Science, Navrongo, Ghana, 4 Department of Biochemistry and Forensic Science, School of Chemical and Biochemical Sciences, CK Tedam University of Technology and Applied Science, Navrongo, Ghana, 5 Faculty of Health Sciences, Department of Chemical Pathology, National Health Laboratory Service, University of the Witwatersrand, Johannesburg, South Africa, 6 African Population and Health Research Center, Nairobi, Kenya, 7 Faculty of Health Sciences, MRC/Wits Developmental Pathways for Health Research Unit, University of the Witwatersrand, Johannesburg, South Africa, 8 Institut de Recherché en Sciences de la Santé, Clinical Research Unit of Nanoro, Clinical Research Unit of Nanoro, Burkina Faso, 9 Faculty of Health Sciences, Sydney Brenner Institute for Molecular Bioscience, University of the Witwatersrand, Johannesburg, Johannesburg, 10 Faculty of Health Sciences, MRC/Wits Rural Public Health and Health Transitions Research Unit (Agincourt), School of Public Health, University of the Witwatersrand, Johannesburg, South Africa, 11 Faculty of Health Sciences, Department of Pathology and Medical Science, DIMAMO, School of Health Care Sciences, University of Limpopo, Polokwane, South Africa, 12 Faculty of Health Sciences, Department of Medicine, Division of Endocrinology & Metabolism, Carbohydrate & Lipid Metabolism Research Unit, Johannesburg Hospital, University of the Witwatersrand, Johannesburg, South Africa

† Deceased.

* drenanonterah@gmail.com (EAN); michele.ramsay@wits.ac.za (MR)

**Data Availability Statement:** All AWI-Gen data can be accessed from the European Genome-phenome Archive (https://ega-archive.org/dacs/EGAC00001000648?order=stable_id&sort=asc).

## Abstract

### Introduction

The contribution of obesity phenotypes to dyslipidaemia in middle-aged adults from four sub-Saharan African (SSA) countries at different stages of the epidemiological transition has not been reported. We characterized lipid levels and investigated their relation with the growing burden of obesity in SSA countries.

### Methods

A cross-sectional study was conducted in Burkina Faso, Ghana, Kenya and South Africa. Participants were middle aged adults, 40–60 years old residing in the study sites for the past 10 years. Age-standardized prevalence and adjusted mean cholesterol, LDL-C, HDL-C, triglycerides and non-HDL-C were estimated using Poisson regression analyses and association of body mass index (BMI), waist circumference (WC) and waist-to-hip ratio (WTHR)

The phenotype data set accession IDs is
EGAD00010001996.

**Funding:** The AWI-Gen Collaborative Centre is
funded by the National Human Genome Research
Institute (NHGRI), Office of the Director (OD),
Eunice Kennedy Shriver National Institute Of Child
Health & Human Development (NICHD), the
National Institute of Environmental Health Sciences
(NIEHS), the Office of AIDS research (OAR) and the
National Institute of Diabetes and Digestive and
Kidney Diseases (NIDDK), of the National Institutes
of Health (NIH) under award number
U54HG006938 and its supplements, as part of the
H3Africa Consortium. The study was also partly
funded the Department of Science and Technology,
South Africa, award number DST/CON 0056/2014,
and by the African Partnership for Chronic Disease
Research (APCDR). The funders had no role in
study design, data collection and analysis, decision
to publish, or preparation of the manuscript.

**Competing interests:** The authors have declared
that no competing interests exist.

with abnormal lipid fractions modeled using a random effects meta-analysis. Obesity phenotypes are defined as BMI $\geq$ 30 kg/m$^2$, increased WC and increased waist-to-hip ratio.

## Results

A sample of 10,700 participants, with 54.7% being women was studied. Southern and Eastern African sites recorded higher age-standardized prevalence of five lipid fractions then West African sites. Men had higher LDL-C (19% vs 8%) and lower HDL-C (35% vs 24%) while women had higher total cholesterol (15% vs 19%), triglycerides (9% vs 10%) and non-HDL-cholesterol (20% vs 26%). All lipid fractions were significantly associated with three obesity phenotypes. Approximately 72% of participants in the sample needed screening for dyslipidaemia with more men than women requiring screening.

## Conclusion

Obesity in all forms may drive a dyslipidaemia epidemic in SSA with men and transitioned societies at a higher risk. Targeted interventions to control the epidemic should focus on health promoting and improved access to screening services.

## Introduction

Excess body fat and its metabolic consequences are recognised global epidemics strongly linked to cardiovascular disease (CVD) morbidity and mortality [1] with 80% of the 18.6 million global CVD-related deaths recorded in 2019 occurring in low- and middle-income countries [2]. CVDs have been increasing in sub-Saharan Africa (SSA) while developed countries have witnessed a steady decline [3,4]. The burden and risk factors of CVDs in SSA differ between countries reflecting the different stages of the epidemiological health transition [5–8].

Central to the epidemiological transition is rising obesity levels, which the Non-Communicable Disease (NCD) Risk Factor Collaboration suggest is driven by rural communities [9,10]. The possible reasons for the rise in rural obesity include increasing incomes, better infrastructure, more mechanized agriculture and increased car use, all of which lead to lower energy expenditure and greater access to energy dense foods [9,10]. The principal metabolic comorbidity associated with excessive body fat is dyslipidaemia. Typically, obesity is strongly associated with atherogenic dyslipidaemia, characterized by high triglyceride (TGs) and low high-density lipoprotein cholesterol (HDL-C) levels [1]. Low-density lipoprotein cholesterol (LDL-C) is an independent predictor of CVD and is the primary target of treatment with non-HDL-C being regarded as a secondary treatment target in the control of dyslipidaemia for the prevention of CVD [1]. While general obesity measured by BMI is a suitable index for evaluating overall adiposity it is a poor indicator of body fat distribution. Central (such as waist circumference) as compared to peripheral body (measured as hip circumference) fat is a major risk factor for coronary artery disease, myocardial infarction and peripheral vascular disease (because it is a metabolically active fat depot and is associated with a higher risk of CVD, according to both epidemiological and Mendelian randomization studies [11,12], which offer both population level observed risk and a genetic-linked risk. Collectively, obesity and central adipose tissue depots contribute to unfavorable levels of dyslipidemia and further promote the development of atherosclerosis and established CVD.

The prevalence of dyslipidaemia including elevated total cholesterol (TC), LDL-C and TGs and low levels of HDL-C [13–15] is increasing in SSA but vary between countries [8,16]. Although ethnic differences in serum lipid levels have been reported [17,18], adiposity in addition to unhealthy diet, low physical activity and poor access to healthcare, especially in low- and middle-income countries are the main drivers of dyslipidaemia [19]. There is the need to create awareness and subsidize screening for lipid abnormalities so as to identify individuals at high risk for developing CVD in the context of rising obesity levels in SSA.

Large-scale population based studies with harmonized data collection of serum lipid levels in Africa are lacking with few studies small scale and localized regional studies [20–22]. To fill these critical gaps, the H3Africa Collaborative Centre, referred to as the Africa Wits-INDEPTH (International Network for the Demographic Evaluation of Populations and Their Health in low- and middle-income countries) partnership for genomics studies (AWI-Gen) generated a cohort of over 10,700 participants from across SSA [23,24]. This study reports burden of abnormal lipid fraction, dyslipidemia and their associations with obesity phenotypes and number needing dyslipidemia screening in middle-aged women and men in four SSA countries in different stages of epidemiological transition.

## Methods

The H3Afric AWI-Gen study received ethical approval from University of Witwatersrand Ethics Committee in South Africa (M121029; M170880) and from each of the participating sites. Written and signed or thump printed informed consent was obtained from all participants before recruitment. This paper follows the Strengthening the Reporting of Observational Studies in Epidemiology (STROBE) reporting guidelines for observational studies. Recruitment of participants across the six sites involved in the study are as follows: Agincourt (13/11/2014 to 30/11/2015); DIMAMO/Digkale (02/11/2014 to 19/08/2016); Nairobi (04/11/2014 to 08/11/2015); Nanoro (20/01/2015 to 25/07/2016); Navrongo (02/02/2015 to 06/10/2015) and Soweto (18/08/2011 to 08/12/2014).

### Study design, setting, population and sampling

A cross-sectional study was conducted in which participants were recruited between 2013 and 2016 in five INDEPTH-Network Health and Demographic Surveillance Sites i.e., Agincourt (rural), Dikgale (rural) in South Africa, Navrongo (rural) in Ghana, Nanoro (rural) in Burkina Faso and Nairobi (urban) in Kenya. The sixth site was the urban Soweto cohort located within the MRC/Wits Developmental Pathways for Health Research Unit (DPHRU) also in South Africa [23–25]. The population was made up of women and men 40-60years, randomly selected from within the study sites. Pregnant women were excluded as well as participants who could not complete the prescribed study procedures. Participants from Nanoro, Nairobi and Navrongo HDSS were selected by simple random sampling using existing sampling frames with an equal number of females and males. In Agincourt the population census was used as the sampling frame in which convenience sampling was used to recruit participants between the ages of 40 and 60 years. In Soweto, men were recruited through simple random sampling from the Soweto community while women were samples from care givers of the "Birth-to-twenty cohort" [26]. Data from 10700 participants was used in this study.

### Sample size determination

Using a pooled prevalence of dyslipidaemia of 16.5% [27] taken from a systematic review of studies in Africa with a margin of error of 5% and a standard deviation of 1.96, a minimum sample size of 1311 was determined using Cochran formula [28]. Therefore, a sample size of

10700 participants used in this study is sufficient to determine the levels of lipid fractions and the prevalence of dyslipidaemia and associated factors in these African populations.

## Data collection

Trained field staff used a standard structured AWI-Gen questionnaire with modifications to suit each country context to collect data. Data were entered or imported into the RedCap electronic database.

## Measures of obesity phenotypes

**BMI in Kg/m$^2$ (an indicator of general obesity).** Standing height was measured using a Harpenden digital stadiometer (Holtain, Crymych, Wales) while weight was measured using digital Physician Large Dial 200kg capacity scales (Kendon Medical, UK). BMI was computed as weight over height in meters squared and classified into four categories; underweight = BMI < 18.5 kg/m$^2$, normal weight = BMI 18.5 kg/m$^2$ to 24.9 kg/m$^2$, overweight weight BMI 25 kg/m$^2$ to 29.9 kg/m$^2$ and obese BMI $\geq$ 30 kg/m$^2$ [29].

**Waist and hip circumference.** Waist circumference was measured in light clothes using a stretch-resistant tape measure (SECA, Hamburg, Germany). Abnormal WC, an indicator of central obesity was defined as waist circumference $\geq$80 cm for women and $\geq$94 cm for men [30]. The hip circumference, as gluteofemoral region, was measured by placing the tape around the most protruding part of the buttocks, ensuring that the zero mark was to the participant's side. The measurement was done to the nearest 0.1cm.

**Waist-to-hip ratio (WTHR).** The ratio of waist and hip circumference was subsequently computed. Abnormal WTHR ratio that can cause substantial risk of cardiometabolic diseases was defined as $\geq$0.90cm for men and $\geq$0.85cm for women.

## Serum lipid fractions and dyslipidemia

Fasting venous blood samples were analyzed at a central laboratory using a Randox Daytona Plus (Randox Laboratories Ltd, UK) autoanalyser. The TC, TG and HDL-C were measured by enzymatic colorimetric methods while LDL-C was calculated using the Friedewald equation [31] and non-HDL-C was computed by subtracting HDL-C from TC [32]. For all assays the coefficient of variation of the technician was low (<2%) and the laboratory performance was acceptable using the Randox International Quality Assessment Scheme. Abnormal lipid levels were defined as follows: TC $\geq$5.0mmol/l (hypercholesterolaemia), LDL-C $\geq$3.0 mmol/l, TG $\geq$1.7 mmol/l, and non-HDL-C >3.4mmol/l and low HDL-C <1.0mmol/l for men and <1.3mmol/l for women [32,33]. Dyslipidaemia was defined as having at least one abnormal level of either of the five lipid fractions or being told by a health professional that they have a high cholesterol level or on treatment foe dyslipidaemia. Treatment for dyslipidaemia was a self-report of the use of lipid-lowering therapy among those who were aware of their condition.

## Other risk factors of CVD

Other variables included risk factors of CVD included demographic (such as, age, sex, education, household socioeconomic status); self-reported behavioral risk (smoking, alcohol use, physical activity, fruit and vegetable intake as self-reported daily number of servings of fruits and vegetables); metabolic risk (menopause status for women, diabetes and hypertension). Details of the definitions of these variables have been published by Ali et al, 2018 [24].

## Statistical analysis

Characteristics of the study participants were summarized using counts and proportions for categorical data and means and standard deviations (±SD) for continuous data due to approximate normal distribution. Age-standardised prevalence rates of abnormal lipid levels and dylipidaemia were calculated using the age distribution of the total AWI-Gen study population and this method has previously been used [34,35]. A mixed effect linear regression with a random effect of mean serum lipid levels predicted by socio-demographic factors, the behavioral risk factors of CVD, and metabolic and anthropometric indices was computed. Post regression estimated adjusted means with standard errors generated through the delta method.

Regional differences in prevalence of dyslipidaemia were computed using Poisson regression analyses with a variance-covariance method to obtain robust standard errors. The effects are presented as prevalence ratios (PRs) with 95% confidence intervals. The PRs indicate the ratio between the prevalence of an outcome in the most transitioned sites versus the least transitioned sites. The populations in West Africa served as the reference group. A random mixed effects logistic regression meta-analysis for the combined AWI-Gen sample to determine association of obesity phenotypes and dyslipidemia as well with each of the lipid fractions. Multiplicative interaction terms between sex and the adiposity measures into each the models were used to determine sex differences.

Finally, the number of participants needing screening for dyslipidemia was computed using World Health Organization package of essential non-communicable disease interventions for primary healthcare in low resource settings (WHO PEN) [36]. This is defined as those exhibiting at least one of the following risk factors: smoking; elevated glucose; high blood pressure; waist circumference ≥90cm in males; waist circumference ≥100cm in females. In the original recommendation age >40 years was recommended but since the minimum age for our study was 40 years, we omitted age from our computation. Statistical significance for all the inferential statistics was set at a p-value of less than 0.05. All analyses were carried out using STATA version 14.2 SE.

## Results

### Participant characteristics

A total of 10,700 participants (55.5% women), average age 50 ± 5 years (women) and 50 ± 6 years (men) from six sites in four SSA countries were studied (Table 1). Current smoking levels were higher in men than women at all sites and were highest in Soweto in both sexes. A high proportion of women and men across all sites were deemed physically active. In both men and women, general, central and peripheral obesity was higher in the South African and East African sites compared to the West African sites and the proportions were higher for women across all sites except in Burkina Faso where the prevalence of obesity was higher in men than women (Table 1).

### Mean serum lipid levels

The adjusted mean levels of the various lipid fractions are presented in Fig 1A for women and 1B for men. The adjusted mean lipid levels differed by study site and sex. There was a general trend of low means lipid levels among women and men from Nanoro and Navrongo (West African sites) except for mean LDL-C levels where men from Nanoro, Burkina Faso and Nairobi, Kenya presented with higher mean levels compared to other sites. South African men had the highest levels of each of the 4 lipid fractions and the lowest HDL-C levels.

**Table 1. Basic characteristics of AWI-Gen participants in six sites from sub-Saharan Africa stratified by sex.**

| African region | South Africa | | | East Africa | West Africa | | |
|---|---|---|---|---|---|---|---|
| Variable | Agincourt (n = 1465) | Dikgale (n = 1212) | Soweto (n = 2030) | Nairobi (n = 1951) | Nanoro (n = 2092) | Navrongo (n = 2014) | All (N = 10700) |
| **Women** | | | | | | | |
| N (%) | 892 (60.9) | 845 (69.7) | 1,003 (49.4) | 1,059 (54.3) | 1,040 (49.7) | 1,091 (54.2) | 5,930 (55.1) |
| Age in years | 50.9±5.8 | 50.5±6.0 | 49.1±5.6 | 48.3±5.3 | 49.8±5.6 | 51.6±5.7 | 50.0±5.8 |
| Formal educational | 612 (68.6) | 737 (90.8) | 784 (7.2) | 1,795 (92.4) | 70 (6.74) | 243 (22.3) | 3,389 (57.5) |
| Household SES | 7 (4–9) | 10 (8–12) | 7 (6–9) | 10 (8–13) | 11 (8–13) | 8 (6–11) | 9 (6–11) |
| Current smoking | 10 (1.12) | 54 (6.65) | 100 (9.98) | 81 (7.67) | 2 (0.19) | 37 (3.39) | 284 (4.82) |
| Current alcohol use | 161 (18.1) | 248 (30.5) | - | 298 (28.2) | 858 (78.6) | 759 (70.1) | 2,324 (39.4) |
| Fruit/vegetable intake | 343 (38.4) | 327 (40.3) | - | 740 (70.1) | 918 (88.4) | 785 (71.9) | 4,115 (69.8) |
| Physically active | 697 (78.6) | 781 (96.3) | 588 (58.7) | 950 (90.2) | 898 (86.4) | 882 (81.1) | 4,796 (81.6) |
| BMI in kg/m$^2$ | 29.2±6.6 | 30.8±8.0 | 33.2±7.2 | 27.6±6.1 | 20.2±3.2 | 22.13.8 | 26.8±7.5 |
| General obesity n, % | 346 (41.8) | 408 (51.2) | 601 (66.3) | 338 (32.2) | 13 (1.3) | 41 (3.8) | 1747 (30.7) |
| WC in cm | 94.8±15.1 | 94.0±16.3 | 98.8±14.3 | 90.6±14.1 | 75.9±8.1 | 76.4±9.3 | 87.9±15.9 |
| WC≥80cm n, % | 693 (82.9) | 622 (78.0) | 818 (90.3) | 804 (76.5) | 254 (24.7) | 324 (30.3) | 818 (90.3) |
| Waist-to-hip ratio | 625 (70.1) | 439 (54.1) | 456 (45.5) | 684 (64.8) | 547 (52.7) | 600 (55.0) | 3,351 (56.9) |
| Post menopause | 537 (36.7) | 558 (47.8) | 668 (33.0) | 599 (30.8) | 717 (34.4) | 718 (35.7) | 3797 (35.5) |
| **Men** | | | | | | | |
| N (%) | 551 (39.7) | 353 (30.7) | 991 (52.2) | 880 (45.6) | 1027 (49.9) | 909 (45.9) | 4711 (45.3) |
| Age in years | 50.8±5.8 | 50.0±6.0 | 49.5±6.0 | 48.8±5.6 | 49.8±6.0 | 50.5±5.7 | 49.8±5.9 |
| Formal educational | 449 (78.5) | 334 (93.8) | 1,017 (99.2) | 852 (96.2) | 283 (27.2) | 351 (38.1) | 3,286 (68.4) |
| Household SES | 6 (4–8) | 9 (7–12) | 12 (10–14) | 11 (9–14) | 11 (9–14) | 9 (7–13) | 10 (8–13) |
| Current smoking | 280 (49.0) | 54 (15.4) | 712 (69.6) | 418 (47.2) | 264 (25.3) | 332 (35.9) | 2,615 (54.5) |
| Current alcohol use | 386 (67.4) | 301 (84.6) | 726 (100) | 629 (71.0) | 768 (73.8) | 850 (92.3) | 3,660 (81.3) |
| Fruit/vegetable intake | 217 (37.9) | 179 (50.3) | - | 584 (65.9) | 854 (81.7) | 672 (72.8) | 3,531 (73.4) |
| Physically active | 444 (77.9) | 340 (96.3) | 840 (81.9) | 846 (95.5) | 782 (74.8) | 814 (89.9) | 4,066 (84.9) |
| BMI in kg/m$^2$ | 24.0±5.2 | 21.7±4.0 | 24.9±5.7 | 22.8±3.9 | 21.6±3.5 | 20.9±3.3 | 22.7±4.6 |
| General obesity | 65 (11.9) | 10 (2.8) | 176 (17.8) | 45 (5.1) | 20 (1.9) | 11 (1.2) | 327 (6.9) |
| WC in cm | 87.0±13.2 | 80.3±11.3 | 89.2±15.0 | 83.4±10.7 | 81.4±9.8 | 73.2±7.4 | 82.4±12.6 |
| WC≥94cm n, % | 151 (27.4) | 46 (13.0) | 361 (36.4) | 146 (16.6) | 114 (11.1) | 17 (1.9) | 835 (17.7) |
| Waist-to-hip ratio | 295 (51.5) | 151 (42.4) | 528 (51.5) | 353 (39.8) | 460 (42.0) | 246 (26.7) | 2,033 (42.3) |

Data presented as absolute count and proportions (%) or mean ±standard deviation (SD) and median (interquartile range) for household socioeconomic status (SES); BMI, body mass index; general obesity is BMI ≥30 kg/m$^2$; WC, waist circumference; central obesity is WC ≥ 80cm in women and WC ≥ 94cm for men; waist-to-hip ratio (WHR) ≥ 0.90 in men and ≥ 0.85; Physically active if they reported a moderate-to-vigorous physical activity of >150 minutes per week.

## Age-standardised prevalence of abnormal lipid levels

The age-standardised prevalence of abnormal lipid levels for each of the lipid fractions and dyslipidaemia in the men and women at the various sites are presented in Table 2. In the combined sample, women were reported a higher prevalence of elevated total cholesterol, triglycerides and non-HDL-C while men had higher prevalence of low HDL-C and elevated LDL-C compared to women. Similarly, women and men from West Africa had the lowest prevalence of elevated TC, LDL-C, TG and non-HDL-C and the lowest prevalence of low HDL-C. Men in Nanoro, Burkina Faso were likely to present with elevated LDL-C compared to Navrongo, Ghana and other sites in South Africa and East Africa. The combined prevalence of dyslipidaemia among women was 58.4% while that among men was 75.4%.

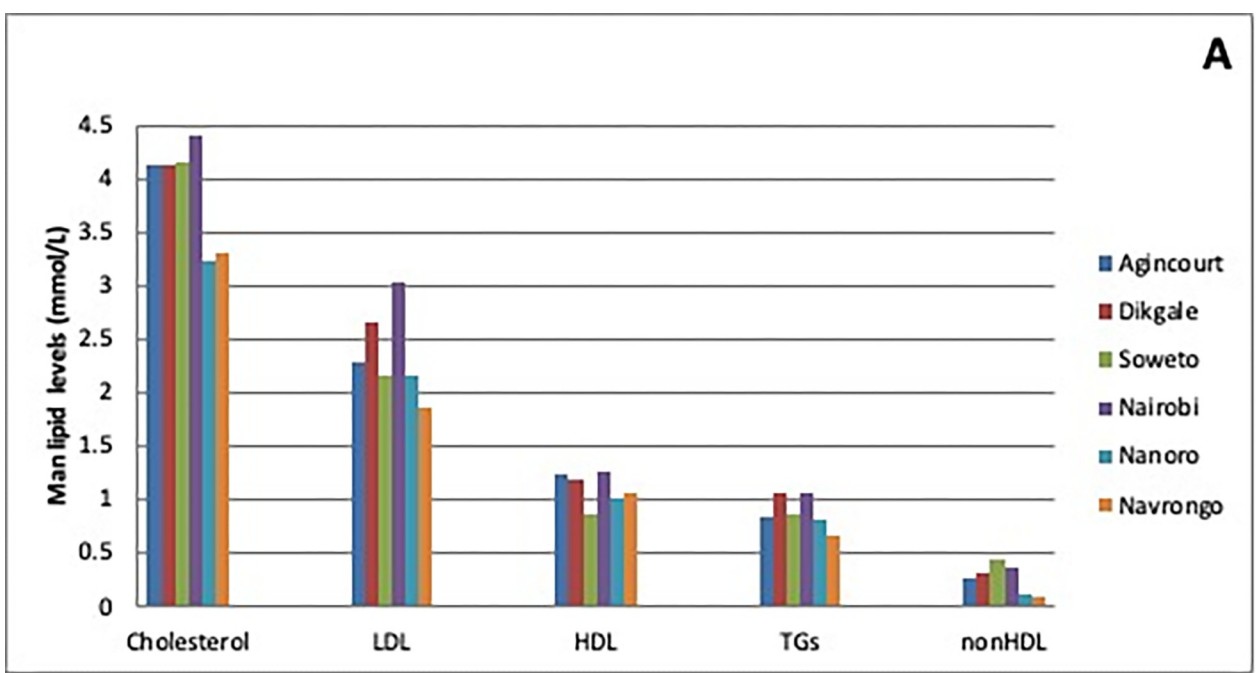

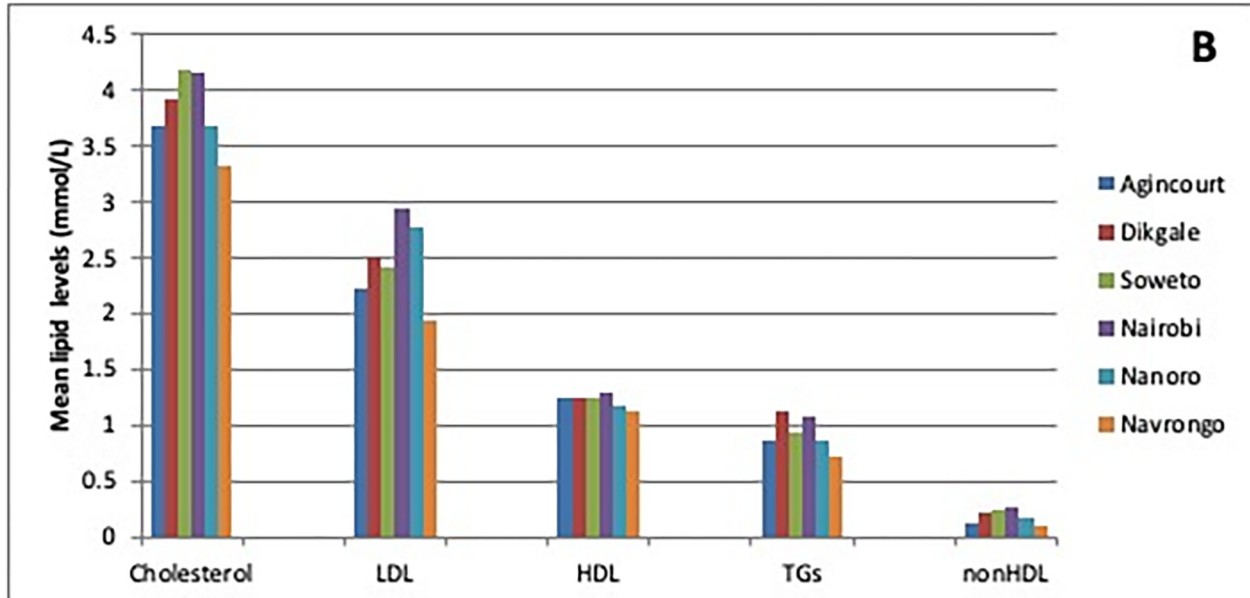

**Fig 1.** Adjusted mean levels of the various lipid fractions in women (A) and men (B) from the AWI-Gen study. Adjusted for age, educational status, household socioeconomic status, smoking, alcohol use, physical activity, fruit and vegetable intake, and use of lipid lowering medication.

### Obesity measures and dyslipidaemia

There were varied associations of BMI, WC and WTHR with dyslipidemia (Table 3). BMI was associated with a greater risk (adjusted odds ratio, AOR [95% confidence interval]) of elevated TC (1.89 [1.57, 2.28]), LDL-C (1.32 [1.06, 1.62]), TG (1.68 [1.32, 2.13]) and non-HDL-C (2.18 [1.84, 2.58]) and a greater low HDL-C (2.18 [1.84, 2.58]) in men and women. In the sex stratified analyses, men had a greater risk of dyslipidaemia than women. Waist circumference was

**Table 2. Age-standardised prevalence rates (with 95% confidence intervals) of elevated lipid fractions among women and men in the six sites of the AWI-Gen study.**

|  | Elevated total cholesterol | Elevated LDL-C | Low HDL-C | Elevated TG | Elevated non-HDL-C | Dyslipidaemia |
|---|---|---|---|---|---|---|
| **Women** |  |  |  |  |  |  |
| Agincourt | 23.4 (20.6–26.1) | 5.27 (3.79–6.75) | 34.1 (30.9–37.2) | 11.3 (9.17–13.3) | 32.7 (29.6–35.7) | 39.5 (32.5–41.5) |
| Dikgale | 22.1 (19.3–24.9) | 8.61 (6.69–10.5) | 29.9 (26.9–33.1) | 13.5 (11.2–15.8) | 32.9 (29.8–36.1) | 34.6 (29.8–37.6) |
| Soweto | 37.3 (34.3–40.3) | 17.5 (15.1–19.1) | 46.1 (42.9–49.2) | 20.1 (17.5–22.6) | 46.3 (43.3–49.4) | 61.2 (60.8–71.4) |
| Nairobi | 27.1 (24.2–30.0) | 10.0 (8.12–11.9) | 37.7 (34.6–40.8) | 12.4 (13.3–14.6) | 38.2 (35.1–41.2) | 26.4 (23.7–37.4) |
| Nanoro | 3.81 (2.6–4.9) | 3.13 (2.08–4.2) | 18.9 (16.5–21.3) | 2.58 (1.60–3.56) | 4.91 (3.6–6.2) | 28.6 (27.1–29.2) |
| Navrongo | 4.53 (3.3–5.8) | 1.80 (1.0–2.6) | 27.3 (24.6–30.1) | 2.97 (1.86–4.1) | 5.77 (4.4–7.1) | 33.5 (31.4–34.8) |
| Combined | 18.8 (17.8–19.8) | 7.7 (7.0–8.4) | 24.3 (20.8–29.6) | 9.93 (9.7–10.7) | 25.7 (24.6–26.8) | 58.4 (48.4–63.4) |
| **Men** |  |  |  |  |  |  |
| Agincourt | 15.0 (12.0–17.9) | 18.1 (14.8–21.3) | 67.2 (63.4–71.1) | 11.0 (8.43–13.7) | 19.3 (16.0–22.5) | 78.7 (66.3–78.9) |
| Dikgale | 14.6 (10.9–18.3) | 18.8 (14.8–22.9) | 70.3 (65.6–75.1) | 12.9 (9.48–16.4) | 22.0 (17.7–26.3) | 80.3 (76.5–90.5) |
| Soweto | 20.1 (17.7–22.6) | 15.3 (13.1–17.5) | 67.7 (64.8–70.5) | 12.5 (10.4–14.5) | 27.3 (24.6–30.1) | 77.8 (69.2–78.4) |
| Nairobi | 22.6 (19.8–25.4) | 31.9 (28.7–35.0) | 68.7 (65.5–71.8) | 12.2 (10.0–14.4) | 29.5 (26.4–32.5) | 86.0 (84.5–89.5) |
| Nanoro | 12.0 (10.1–14.0) | 25.3 (22.7–27.9) | 68.2 (65.4–71.0) | 7.37 (5.78–8.95) | 17.1 (14.8–19.4) | 78.7 (76.7–80.3) |
| Navrongo | 3.76 (2.51–5.0) | 2.77 (1.71–3.8) | 64.8 (61.7–67.9) | 2.15 (1.21–3.1) | 5.40 (3.92–6.9) | 66.3 (86.5–74.3) |
| Combined | 14.7 (13.7–15.7) | 18.7 (17.5–19.7) | 67.8 (53.7–76.3) | 9.23 (8.41–10.5) | 19.9 (18.8–21.1) | 75.7 (64.3–79.3) |

Presented as proportions with 95% confidence intervals; LDL-C calculated using the Friedewald method; LDL-C, low density lipoprotein, HDL-C, high density lipoprotein, TGs, triglycerides; Elevated cholesterol is defined as TC $\geq$ 5 mmol/L; Elevated LDL-C is LDL-C $\geq$3 mmol/L; Low HDL-C is HDL-C <1.0 mmol/L in men and <1.3 mmol/L in women; elevated TGs is TGs $\geq$1.7 mmol/L and elevated non-HDL-C is non-HDL-C >3.4 mmol/L.

associated with elevated TC (1.64 [1.25, 2.14]), LDL-C (1.26 [1.03, 1.54]), TG (1.83 [1.30, 2.57]) and non-HDLC (1.67 [1.32, 2.11]) and a greater low HDL-C (1.12 [1.08, 1.18]). There was a significant sex interaction with all five lipid fractions (p<0.05). Women were more at

**Table 3. Sex differences in the association of the various adiposity phenotypes with the lipid fractions in the combined AWI-Gen cohort.**

| Adiposity phenotype | Sample | Elevated total cholesterol | Elevated LDL-C | Low HDL-C | Elevated TGs | Elevated non-HDL-C |
|---|---|---|---|---|---|---|
| BMI in kg/m$^2$ | *Combine sample* | 1.89 (1.57, 2.28) | 1.32 (1.06, 1.62) | 1.21 (1.16, 1.28) | 1.68 (1.32, 2.13) | 2.18 (1.84, 2.58) |
|  | $P_{Sex*BMI}$ | 0.027 | <0.001 | <0.001 | 0.088 | 0.014 |
|  | Women | 1.74 (1.36, 2.25) | 1.94 (1.33, 2.82) | 1.41 (1.30, 1.56) | 1.75 (1.24, 2.46) | 2.06 (1.65, 2.57) |
|  | Men | 2.15 (1.61, 2.89) | 2.36 (1.78, 3.14) | 1.45 (1.34, 1.62) | 1.84 (1.26, 2.68) | 2.51 (1.88, 3.34) |
| WC in cm | *Combine sample* | 1.64 (1.25, 2.14) | 1.26 (1.03, 1.54) | 1.12 (1.08, 1.18) | 1.83 (1.30, 2.57) | 1.67 (1.32. 2.11) |
|  | $P_{Sex*WC}$ | 0.037 | <0.001 | <0.001 | 0.001 | 0.001 |
|  | Women | 1.96 (1.40, 2.76) | 2.89 (1.87, 4.42) | 1.59 (1.37, 1.94) | 3.04 (1.98, 4.78) | 2.44 (1.82, 3.28) |
|  | Men | 1.67 (1.03, 2.72) | 3.98 (2.91, 5.44) | 1.52 (1.24, 1.89) | 1.69 (0.99, 2.88) | 1.42 (1.09, 2.18) |
| WTHR | *Combine sample* | 1.47 (1.24, 1.75) | 1.77 (1.46, 2.15) | 1.16 (1.04, 1.20) | 2.21 (1.75, 2.79) | 1.63 (1.39, 1.91) |
|  | $P_{Sex*WTHR}$ | 0.007 | <0.001 | <0.001 | 0.097 | <0.001 |
|  | Women | 1.29 (1.04, 1.63) | 1.95 (1.37, 2.78) | 1.18 (1.06, 1.29) | 2.13 (1.56, 2.91) | 1.51 (1.24, 1.85) |
|  | Men | 1.75 (1.32, 2.32) | 1.89 (1.46, 2.44) | 1.17 (1.03, 1.27) | 2.37 (1.65, 3.42) | 1.84 (1.42, 2.37) |

Elevated cholesterol is defined as TC $\geq$ 5 mmol/L; Elevated LDL-C is LDL-C $\geq$3 mmol/L; Low HDL is HDL-C <1.0 mmol/L in men and <1.3 mmol/L in women; elevated TGs is TGs $\geq$1.7 mmol/L and elevated non-HDL-C is non-HDL-C >3.4 mmol/L; results are presented as odds ratios with corresponding 95% confidence intervals; $P_{sex\ interaction}$ represent sex differences derived from sex and adiposity phenotype multiplicative interaction term; the models are adjusted for age, educational status, household socioeconomic status, smoking, alcohol intake, physical inactivity, fruits and vegetable intake and use of lipid lowering medication; *The BMI model had only WTHR it them will BMI was included in WC and WTHR model.

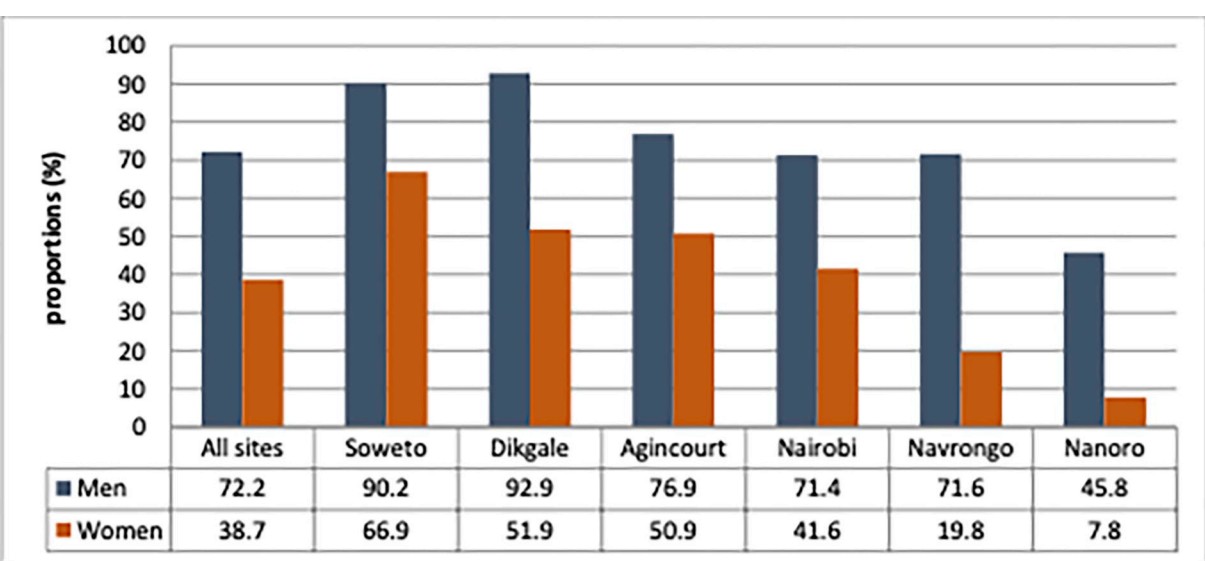

**Fig 2. The total number of participants recommended for dyslipidaemia screening computed using the protocol of World Health Organization Package of Essential Non communicable disease intervention for primary healthcare in low resource settings (WHO PEN).**

risk of all abnormal lipid fractions but LDL-C where men had a greater risk than women. Waist-to-hip ratio was associated with a greater odds of elevated TC (1.47 [1.25, 1.75]), LDL-C (1.77 [1.46, 2.15]), Triglycerides (2.21 [1.75, 2.79]) and non-HDLC (1.63 [1.39, 1.91]) and a greater odds of low HDL-C (1.16 [1.04, 1.20]) differences between women and men except for Triglycerides.

### Regional difference

We first reported association of each phenotype with the lipid fractions for the sub-regional blocks (S1 Table) and further examined regional differences in prevalence ratio of the abnormal fractions with West Africa as a reference (S2 Table). There was an observed gradient regarding the prevalence ratio of all five lipid fractions with East Africa and South Africa having the higher prevalence ratio compared to West Africa. Similarly, the association of obesity phenotypes with dyslipidemia showed higher odds in Southern and east Africa compared to West Africa.

### Number recommended for screening

The number of participants needing dyslipidemia screening, 72.2% of men and 38.7% of women in the total AWI-Gen population would benefit from lipid screening. Sites in South Africa had a higher need for screening compared to East and West African sites (Fig 2).

### Discussion

In this large multi-county SSA study we observed that the mean lipid levels were higher in Southern (Agincourt, Dikgale and Soweto) and Eastern Africa (Nairobi, Kenya) compared to the two sites in West Africa (Nanoro, Burkina Faso and Navrongo, Ghana). We also observed that the age-standardized prevalence of dyslipidaemia followed a similar pattern. We further observed that all forms of obesity were associated with dyslipidaemia in the full cohort and in the regional blocks. These associations had a similar direction of but the magnitude of association was higher in South Africa then East compared to West Africa. Finally, a greater

proportion of the AWI-Gen participants needed screening for dyslipidemia with more men than women needing screening.

The observed rural-urban gradient in the burden of dyslipidaemia in our study has been previously reported. The Research on Obesity and Diabetes among African Migrants (RODAM) study previously reported Ghanaian men living in rural Ghana had a lower burden of high LDL-C, high TC levels and elevated TGs compared to those in urban Ghana and Ghanaian migrants in Europe [34,35]. Other studies within the African continent have also reported on rural-urban difference in dyslipidaemia [28,36–38] with a systematic review and meta-analysis, reporting that East and Southern African countries were likely to present with a higher prevalence of elevated LDL-C compared to West African countries [39]. Epidemiological data on morbidity and mortality due to CVD and data on CMD risk factors from the current and a previous study [40,41] suggest that the West African sites are the least transitioned with East Africa and South African sites being further along the transition pathway. In addition, multiple studies conducted at each site have allowed in-depth characterisation of the prevailing sociodemographic features and demonstrate that both West African sites are rural with residents reliant on subsistence farming, whilst the site in Nairobi is an urban shanty town, and the sites in South Africa are a mix of peri-urban (Dikgale and Agincourt) and urban (Soweto). Based on these characteristics and the findings, we suggest the influence of epidemiological transition on dyslipidaemia with the two West African populations depicting early stage of epidemiological health transition compared to Nairobi (East Africa) and South African populations.

Despite this, some variations were observed in the prevalence of abnormal levels of LDL-C and HDL-C. For instance, men the West African site of Nanoro (68%) had a higher prevalence of low HDL-C than Navrongo, Ghana (64%) but similar to East Africa (Nairobi, 68%). The RODAM study previously made similar observations (23). Africans in general present with more favorable lipid profiles compared to European and other race-ethnic groups [42,43] but evidence from African American populations demonstrate that this does not translate into lower cardiovascular events [44]. Recent genome-wide association (GWAS) meta-analysis of the AWI-Gen cohort with four other African cohorts suggest genetic and environmental contribution to variation in lipid levels across SSA and between African and non-African population [45], but further studies are required to confirm these observations as well to investigate the contribution of low HDL-C to CVD risk.

Women were likely to present with a high prevalence of all lipid abnormalities except elevated LDL-C. There are inconsistencies in the literature regarding this finding with some studies reporting similar data to our study [42] while other studies demonstrated a higher burden of dyslipidaemia in men [43,44]. Differences in lipid metabolism between men and women have been reported with women of child bearing age likely to have high atheroprotective lipids (HDL-C) compared to men with the reverse observed during postmenopausal stage [44]. Further studies establishing the causal link between genetic, metabolic (including sex hormones) and lifestyle-associated predisposition to elevated LDL-C will be useful in explaining the observed sex differences.

In a systematic review, independent predictors of dyslipidemia in different African settings included high BMI and waist circumference [45]. We observed that high BMI, waist circumference and WTHR, were associated with a higher odd of dyslipidaemia. The adipose tissue that contributes to central obesity (of which waist circumference and WTHR are proxy indicators) includes visceral and subcutaneous fat. The visceral fat component of waist circumference is a metabolically active endocrine and immune organ and correlates more strongly with CMD risk factors than does subcutaneous fat due to its higher output of cytokines [46] and its release of FFAs into the portal circulation A previous study using the AWI-Gen cohort had

demonstrated that adiposity phenotypes such as general obesity (BMI), central obesity (waist circumference and visceral fat) were associated with subclinical atherosclerosis in African populations [47].

Consistent with our findings, a previous study had observed a high unmet need for the identification and treatment of hypercholesterolemia in 35 low- and middle-income countries [48] while another study also observed low awareness and poor control of dyslipidaemia in Africa [49]. There is a greater need for systematic screening to detect dyslipidaemia at the early stages, education and prompt management within the population and through primary health care strengthening such as provision of point-of-care testing methods. With effective screening, minor issues could be identified in a timely manner to enable implementation of preventive measures that will prevent complications or overt clinical conditions. This could further enable the health system to assess the effectiveness of interventions and monitor treatment outcomes.

## Strengths and limitations

The availability of data from three different geographical regions in SSA is a unique strength of the AWI-Gen study. It provided us the opportunity to report data from countries representing different stages of the epidemiological transition using three sub-regional blocks in SSA. Another major strength of this study is the use of highly standardised procedures in data collection across all study sites as well as centralised analyses of serum lipid levels minimising measurement variability. The absence of wide confidence intervals further confirms that major within and across sites variations were minimized. Given the cross-sectional nature of this study, causality cannot be demonstrated and residual confounding from other unmeasured variables cannot be ruled out.

## Conclusion

Obesity in all forms may drive a dyslipidaemia epidemic in sub-Saharan Africa with men and transitioned societies at higher risk. Interventions should aim at reducing the burden of obesity in SSA countries with attention paid to dietary intake, physical activity, improved access to screening services at the primary healthcare level. Additional research is needed to establish the true contribution of low HDL-C to CVD risk in African populations.

## Supporting information

**S1 Checklist. STROBE statement—checklist of items that should be included in reports of observational studies.**
(DOC)

**S1 Table. Regional differences in the association of the various adiposity phenotypes with abnormal lipid fractions in the combined AWI-Gen cohort.** Elevated cholesterol is defined as TC ≥ 5 mmol/L; Elevated LDL-C is LDL-C ≥3 mmol/L; Low HDL is HDL-C <1.0 mmol/L in men and <1.3 mmol/L in women; elevated TGs is TGs ≥1.7 mmol/L and elevated non-HDL-C is non-HDL-C >3.4 mmol/L; results are presented as odds ratios with corresponding 95% confidence intervals; the models are adjusted for age, educational status, household socio-economic status, smoking, alcohol intake, physical inactivity, fruits and vegetable intake and use of lipid lowering medication; *The BMI model had only WTHR it them will BMI was included in WC and WTHR model.
(DOCX)

**S2 Table. Prevalence ratio of abnormal lipid levels sub-regional blocks with West Africa as the reference.** Data presented as prevalence ratios with West Africa as the reference group. West Africa includes Nanoro, Burkina Faso and Navrongo, Ghana sites; East Africa includes Nairobi, Kenya; and South Africa, Agincourt, Dikgale and Soweto sites; LDL-C, low density lipoprotein cholesterol; HDL-C, high density lipoprotein cholesterol and non-HDL-C, non-high-density lipoprotein cholesterol.
(DOCX)

**S3 Table. STROBE statement—checklist of items that should be included in reports of observational studies.**
(DOC)

**S1 File.**
(DOCX)

## Acknowledgments

This paper is dedicated to the memory of Professor Marianne Alberts formerly head of the Dikgale (later renamed DIMAMO) health and demographic surveillance site in Limpopo, South Africa. She sadly passed away before submission of this manuscript. This study would not have been possible without the generosity of the participants who spent many hours responding to questionnaires, being measured, and having samples taken. We wish to acknowledge the sterling contributions of our field workers, phlebotomists, laboratory scientists, administrators, data personnel and other staff who contributed to the data and sample collections, processing, storage, and shipping. We would like to acknowledge investigators from the various sites for the significant contributions made to this research.

AWI-Gen and the H3Africa Consortium: Lucas Amenga-Etego[1], Cornelius Debpuur[1], Eric Fato[1], Immaculate Anati[1] ([1]Navrongo Health Research Centre, Ghana Health Service, Navrongo, Ghana); Christopher Khayeka–Wandabwa[6], Tilahun Nigatu Haregu[6], Stella Muthuri[6] (African Population Health Research Center, Nairobi, Kenya); Nomses Baloyi[7], Juliana Kagrana[7], Richard Munthali[7], Yusuf Guman[7]; Toussaint Rouamba[8], Seydou Diallo-Nakanabo[8] ([8]Clinical Research Unit of Nanoro, Institut de Recherché en Sciences de la Santé, Clinical Research Unit of Nanoro, Burkina Faso); Freedom Mukomana[9], Ananyo Choudary[9], Zane Lombard[9], Scot Hezelhurst[9] ([9]Sydney Brenner Institute for Molecular Bioscience, Faculty of Health Sciences, University of the Witwatersrand, Johannesburg); Francesc Xavier Gomez-Olive Casas[10], Kathleen Kahn[10] ([10]MRC/Wits Rural Public Health and Health Transitions Research Unit (Agincourt), School of Public Health, Faculty of Health Sciences, University of the Witwatersrand, Johannesburg 2193, South Africa); Given Mashaba[11], Felistas Mashinya[11] and Sam Ntuli[11] ([11]DIMAMO, Department of Pathology and Medical Science, School of Health Care Sciences, Faculty of Health Sciences, University of Limpopo, Polokwane, South Africa).

## Author Contributions

**Conceptualization:** Engelbert A. Nonterah, Godfred Agongo, Nigel J. Crowther, Shukri F. Mohamed, Lisa K. Micklesfield, Palwendé Romuald Boua, Shane A. Norris, Stephen M. Tollman, Frederick J. Raal, Marianne Alberts, Michele Ramsay.

**Data curation:** Engelbert A. Nonterah, Godfred Agongo, Shukri F. Mohamed, Palwendé Romuald Boua, Solomon S. R. Choma, Hermann Sorgho, Isaac Kissiangani, Michele Ramsay.

**Formal analysis:** Engelbert A. Nonterah.

**Funding acquisition:** Engelbert A. Nonterah, Nigel J. Crowther, Stephen M. Tollman, Marianne Alberts, Michele Ramsay.

**Investigation:** Engelbert A. Nonterah, Godfred Agongo, Shukri F. Mohamed, Palwendé Romuald Boua, Alisha N. Wade, Solomon S. R. Choma, Hermann Sorgho, Isaac Kissiangani, Gershim Asiki, Shane A. Norris, Stephen M. Tollman, Frederick J. Raal, Marianne Alberts, Michele Ramsay.

**Methodology:** Engelbert A. Nonterah, Nigel J. Crowther, Shukri F. Mohamed, Palwendé Romuald Boua, Alisha N. Wade, Solomon S. R. Choma, Hermann Sorgho, Gershim Asiki, Shane A. Norris, Stephen M. Tollman, Frederick J. Raal, Marianne Alberts, Michele Ramsay.

**Project administration:** Nigel J. Crowther, Solomon S. R. Choma, Hermann Sorgho, Gershim Asiki, Abraham R. Oduro, Shane A. Norris, Stephen M. Tollman, Marianne Alberts, Michele Ramsay.

**Resources:** Stephen M. Tollman, Marianne Alberts, Michele Ramsay.

**Software:** Engelbert A. Nonterah.

**Supervision:** Nigel J. Crowther, Shane A. Norris, Stephen M. Tollman, Frederick J. Raal, Marianne Alberts, Michele Ramsay.

**Validation:** Engelbert A. Nonterah, Godfred Agongo, Nigel J. Crowther, Shukri F. Mohamed, Lisa K. Micklesfield, Palwendé Romuald Boua, Alisha N. Wade, Solomon S. R. Choma, Hermann Sorgho, Isaac Kissiangani, Gershim Asiki, Patrick Ansah, Abraham R. Oduro, Shane A. Norris, Stephen M. Tollman, Frederick J. Raal, Marianne Alberts, Michele Ramsay.

**Visualization:** Engelbert A. Nonterah, Godfred Agongo, Nigel J. Crowther, Shukri F. Mohamed, Lisa K. Micklesfield, Palwendé Romuald Boua, Alisha N. Wade, Solomon S. R. Choma, Hermann Sorgho, Isaac Kissiangani, Gershim Asiki, Patrick Ansah, Abraham R. Oduro, Shane A. Norris, Stephen M. Tollman, Frederick J. Raal, Marianne Alberts, Michele Ramsay.

**Writing – original draft:** Engelbert A. Nonterah, Shukri F. Mohamed, Palwendé Romuald Boua, Marianne Alberts.

**Writing – review & editing:** Engelbert A. Nonterah, Godfred Agongo, Nigel J. Crowther, Shukri F. Mohamed, Lisa K. Micklesfield, Palwendé Romuald Boua, Alisha N. Wade, Solomon S. R. Choma, Hermann Sorgho, Isaac Kissiangani, Gershim Asiki, Patrick Ansah, Abraham R. Oduro, Shane A. Norris, Stephen M. Tollman, Frederick J. Raal, Marianne Alberts, Michele Ramsay.

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
