## [Decision Letter · Decision Letter 0]

4 Nov 2024

PONE-D-24-06681Obesity phenotypes and dyslipidemia in adults from four African countries: An H3Africa AWI-Gen StudyPLOS ONE

Dear Dr. Nonterah,

Thank you for submitting your manuscript to PLOS ONE. After careful consideration, we feel that it has merit but does not fully meet PLOS ONE’s publication criteria as it currently stands. Therefore, we invite you to submit a revised version of the manuscript that addresses the points raised during the review process.

This is a well-written manuscript that sought to report the burden of abnormal lipid fraction, dyslipidemia and their associations with obesity phenotypes stratified by sex among four SSA countries. I agree with two external referees that the manuscript has some minor issues that could be adressed within a revised version of the manuscript.

We look forward to receiving your revised manuscript.

Kind regards,

Neftali Eduardo Antonio-Villa, MD PhD

Academic Editor

PLOS ONE

Journal Requirements:

"The AWI-Gen Collaborative Centre is funded by the National Human Genome Research Institute (NHGRI), Office of the Director (OD), Eunice Kennedy Shriver National Institute Of Child Health & Human Development (NICHD), the National Institute of Environmental Health Sciences (NIEHS), the Office of AIDS research (OAR) and the National Institute of Diabetes and Digestive and Kidney Diseases (NIDDK), of the National Institutes of Health (NIH) under award number U54HG006938 and its supplements, as part of the H3Africa Consortium. The study was also partly funded the Department of Science and Technology, South Africa, award number DST/CON 0056/2014, and by the African Partnership for Chronic Disease Research (APCDR)."

5. Please note that your Data Availability Statement is currently missing the repository name and/or the DOI/accession number of each dataset OR a direct link to access each database. If your manuscript is accepted for publication, you will be asked to provide these details on a very short timeline. We therefore suggest that you provide this information now, though we will not hold up the peer review process if you are unable.

6. One of the noted authors is a group or consortium [Freedom Mukomana, Annoy Choudary, Juliana Kagura, Zane Lombard, Scot Hezelhurst]. In addition to naming the author group, please list the individual authors and affiliations within this group in the acknowledgments section of your manuscript. Please also indicate clearly a lead author for this group along with a contact email address.

Additional Editor Comments:

Please, find attach some additional comments:

Abstract

* Adjust the abstract headings to align with the Plos One format.

* Remove the sentence: “(odds ratio [95% confidence interval])” from the Findings section.

* Clarify that obesity phenotypes are defined as BMI >30, increased waist-to-hip circumference, and increased waist-to-hip ratio.

Methods and Results

* Include the STROBE checklist as a supplementary material file.

* Although the H3Africa AWI-Gen study is a previously published paper, specify the sampling method used and clarify if population weights were applied.

* Indicate whether “behavioral risk factors” were self-reported by the study participants.

* In the “mixed-effect linear regression” section, specify the random effect used in the model.

* Correct the typo in Figure 1, y-label of panel A, which should read “Mean lipid Levels.” Also, consider adding the 95% CI for the adjusted mean levels.

* Estimating the age-standardized prevalence rates of any dyslipidemia in the sample would be insightful.

* Review for typos, especially where “non-HLD-C” is written as “non-HLDC.”

Reviewers' comments:

Reviewer's Responses to Questions

**Comments to the Author**

1. Is the manuscript technically sound, and do the data support the conclusions?

Reviewer #1: Yes

Reviewer #2: Yes

2. Has the statistical analysis been performed appropriately and rigorously? 

Reviewer #1: Yes

Reviewer #2: Yes

3. Have the authors made all data underlying the findings in their manuscript fully available?

Reviewer #1: No

Reviewer #2: Yes

4. Is the manuscript presented in an intelligible fashion and written in standard English?

Reviewer #1: Yes

Reviewer #2: No

5. Review Comments to the Author

Reviewer #1: This is a well written manuscript; I have not seen anything untoward. It is for this reason that I am accepting it without any revisions. The structure of the manuscript is well written and adhering to PLOS one requirement.

Reviewer #2: Summary

This study provides current estimates of the prevalence of dyslipidemia in several urban and rural African settings, as well as the associations of dyslipidemia and obesity phenotypes.

General comments

• Check spelling and grammar throughout the manuscript; there were many typos and sentence structure issues scattered throughout the sections.

Introduction

• Page 4, paragraph 2, sentence 1: Suggest adding some more context to why obesity is rising in rural areas, especially as you have several rural study sites.

• Page 4, paragraph 2, sentence 6: What does 120 mean?

• Page 4, paragraph 2, sentence 7: If you distinguish between the two study approaches (epidemiologic versus Mendelian randomization), I suggest adding some context for why both types are needed.

• Page 4, paragraph 3, sentences 1-2: What are the costs associated with dyslipidemia, e.g., healthcare utilization, morbidity, mortality, etc. These could be important reasons for screening, too.

Methods

• Page 6, paragraph 4, sentence 2: What is the rationale for those specific cutoffs?

• Page 8: paragraph 1, sentence 3: Do you mean least transitioned instead of “least exposed?”

• Page 8, paragraph 1, sentence 4: Can you provide some rationale for using the West African sites as the reference, if they are classified as least transitioned?

• Page 8, paragraph 2, sentence 1: Please provide more information on the WHO package, such as a general description of how the statistics are calculated and the criteria for needing screening.

Results

• Table 1: You may want to only report the proportions, not the counts, since those are more difficult to compare across study sites; plus, the number of participants per study site is already reported in the first table row.

• Table 1: How is fruit and vegetable intake measured?

• Table 2: Check the number of decimal places of Nanoro and Navrongo compared to the other study sites.

• Table 2: Double check the combined prevalence rate of low HDL-C in men.

• Table 2: Check the formatting for confidence intervals.

Discussion

• Page 17, paragraph 1, sentence 2: Consider adding information about how the urban Ghanaian population that was also studied.

• Page 17, paragraph 1, sentence 5: What are those studies that describe the sociodemographic characteristics of those study sites? Please cite them.

• Page 17, paragraph 2, sentence 2: Check the reference to the RODAM study; I think there is an incorrect citation. Also, please clarify by adding more description to observations you are discussing.

• Page 18, paragraph 2: While all the measures were associated with higher odds of dyslipidaemia, there were some stark differences in the magnitudes of association. Was this consistent with the results described in the systematic review?

• Page 18, paragraph 3: You may want to also discuss the benefits of better screening on cardiovascular disease in Africa; i.e., what could be achieved by scaling up screening to the levels you ascertained in that part of the analysis?

6. PLOS authors have the option to publish the peer review history of their article (what does this mean?). If published, this will include your full peer review and any attached files.

Reviewer #1: No

Reviewer #2: No

---

## [Author Response · Author response to Decision Letter 0]

2 Dec 2024

Comments Response

Editors Please ensure that your manuscript meets PLOS ONE's style requirements, including those for file naming. We have done this

 Please include a complete copy of PLOS’ questionnaire on inclusivity in global research in your revised manuscript. We have attached this as “supporting information”

 We note that the grant information you provided in the ‘Funding Information’ and ‘Financial Disclosure’ sections do not match. When you resubmit, please ensure that you provide the correct grant numbers for the awards you received for your study in the ‘Funding Information’ section. We have corrected this

Please include this amended Role of Funder statement in your cover letter; we will change the online submission form on your behalf. We had indicated this in the funding section. We have amended the statement to reflect the suggested wording

 Please note that your Data Availability Statement is currently missing the repository name and/or the DOI/accession number of each dataset OR a direct link to access each database. If your manuscript is accepted for publication, you will be asked to provide these details on a very short timeline. We therefore suggest that you provide this information now, though we will not hold up the peer review process if you are unable. We have now provided the data repository name, link and access number: “The AWI-Gen data used in the current study can be accessed from the European Genome-Phenome Archive (https://ega-archive.org/datasets) with phenotype dataset accession ID of EGA00001002482.”

 One of the noted authors is a group or consortium [Freedom Mukomana, Annoy Choudary, Juliana Kagura, Zane Lombard, Scot Hezelhurst]. In addition to naming the author group, please list the individual authors and affiliations within this group in the acknowledgments section of your manuscript. Please also indicate clearly a lead author for this group along with a contact email address. We amended this to include the list of the group authors in the acknowledgement section as requested

 Please review your reference list to ensure that it is complete and correct. If you have cited papers that have been retracted, please include the rationale for doing so in the manuscript text, or remove these references and replace them with relevant current references. Any changes to the reference list should be mentioned in the rebuttal letter that accompanies your revised manuscript. If you need to cite a retracted article, indicate the article’s retracted status in the References list and also include a citation and full reference for the retraction notice. We have revised this accordingly

 Abstract

* Adjust the abstract headings to align with the Plos One format.

* Remove the sentence: “(odds ratio [95% confidence interval])” from the Findings section.

* Clarify that obesity phenotypes are defined as BMI >30, increased waist-to-hip circumference, and increased waist-to-hip ratio. *We have adjusted the heading (page 3)

*The suggested sentence has been removed (page 3) and *we have clarified as suggested (refer to page 3)

 Methods and Results

* Include the STROBE checklist as a supplementary material file.

* Although the H3Africa AWI-Gen study is a previously published paper, specify the sampling method used and clarify if population weights were applied.

* Indicate whether “behavioral risk factors” were self-reported by the study participants.

* In the “mixed-effect linear regression” section, specify the random effect used in the model.

* Correct the typo in Figure 1, y-label of panel A, which should read “Mean lipid Levels.” Also, consider adding the 95% CI for the adjusted mean levels.

* Estimating the age-standardized prevalence rates of any dyslipidemia in the sample would be insightful.

* Review for typos, especially where “non-HLD-C” is written as “non-HLDC.” We have included the *STROBE checklist as supplementary material now

*We have included a summary statement on sampling methods used in methods section (page 6)

*Behavioural risk was self-reported and this has been indicated now (page 8).

*We have corrected the label for the y-axis now (see page 12)

*We have now specified the random effects used in the models (page 9). 

*We have computed the standardized prevalence of dyslipidaemia now (see 13) and results in page 14 table 2 

* We have reviewed these typos

Reviewer #1 This is a well written manuscript; I have not seen anything untoward. It is for this reason that I am accepting it without any revisions. The structure of the manuscript is well written and adhering to PLOS one requirement. Thank you for the kind words. We appreciate your appraisal of the manuscript 

Reviewer #2 General comments

• Check spelling and grammar throughout the manuscript; there were many typos and sentence structure issues scattered throughout the sections. We have tried to correct any grammatical errors and typos throughout the manuscript

Introduction 

 • Page 4, paragraph 2, sentence 1: Suggest adding some more context to why obesity is rising in rural areas, especially as you have several rural study sites.

• Page 4, paragraph 2, sentence 6: What does 120 mean?

• Page 4, paragraph 2, sentence 7: If you distinguish between the two study approaches (epidemiologic versus Mendelian randomization), I suggest adding some context for why both types are needed.

• Page 4, paragraph 3, sentences 1-2: What are the costs associated with dyslipidemia, e.g., healthcare utilization, morbidity, mortality, etc. These could be important reasons for screening, too. *We have added more to the reasons driving this phenomenon (page 4)

*We have deleted it as we realized it is an error (page 4)

*A phrase has been added to clarify (page 4)

*We have revised this for clarity and included suggestions by the reviewer (page 5)

Methods • Page 6, paragraph 4, sentence 2: What is the rationale for those specific cutoffs?

• Page 8: paragraph 1, sentence 3: Do you mean least transitioned instead of “least exposed?”

• Page 8, paragraph 1, sentence 4: Can you provide some rationale for using the West African sites as the reference, if they are classified as least transitioned?

• Page 8, paragraph 2, sentence 1: Please provide more information on the WHO package, such as a general description of how the statistics are calculated and the criteria for needing screening. *There are no specific cutoffs for Africa hence we used WHO recommendations

*We have corrected to “least transitioned” (page 8)

*In comparative analyses the interest is risk hence it is assumed least transitioned of the sites will have lower risk.

*We added a phrase to describe this (page 9) 

Results • Table 1: You may want to only report the proportions, not the counts, since those are more difficult to compare across study sites; plus, the number of participants per study site is already reported in the first table row.

• Table 1: How is fruit and vegetable intake measured?

• Table 2: Check the number of decimal places of Nanoro and Navrongo compared to the other study sites.

• Table 2: Double check the combined prevalence rate of low HDL-C in men.

• Table 2: Check the formatting for confidence intervals. *We appreciate the comment and have thus edited the table as requested

*We have included this in the methods now (page 7&8)

*Number of decimal places have been corrected now to be consistent

*The combined prevalence rate has been amended after rerunning the analyses

*Confidence intervals have been reformatted and are now consistent 

Discussion • Page 17, paragraph 1, sentence 2: Consider adding information about how the urban Ghanaian population that was also studied.

• Page 17, paragraph 1, sentence 5: What are those studies that describe the sociodemographic characteristics of those study sites? Please cite them.

• Page 17, paragraph 2, sentence 2: Check the reference to the RODAM study; I think there is an incorrect citation. Also, please clarify by adding more description to observations you are discussing.

• Page 18, paragraph 2: While all the measures were associated with higher odds of dyslipidaemia, there were some stark differences in the magnitudes of association. Was this consistent with the results described in the systematic review?

• Page 18, paragraph 3: You may want to also discuss the benefits of better screening on cardiovascular disease in Africa; i.e., what could be achieved by scaling up screening to the levels you ascertained in that part of the analysis? *We did not study an urban Ghanaian population

*The first paragraph was a summary of our results hence we didn’t cite. 

*We have corrected the citation for the RODAM study and we have added more to the observations reported (page 18)

*Yes, and we have revised for clarity (page 20 lines 549-552)

The magnitudes of the association in the systematic review are consistent with our findings where the odds of dyslipidaemia are higher in East and Southern Africa compared to those in West Africa.

*We have stated the benefit of better screening on cardiovascular disease in Africa (see page 20, lines 549-552)

---

## [Editor Report · Decision Letter 1]

13 Dec 2024

Obesity phenotypes and dyslipidemia in adults from four African countries: An H3Africa AWI-Gen Study

PONE-D-24-06681R1

Dear Dr. Nonterah,

We’re pleased to inform you that your manuscript has been judged scientifically suitable for publication and will be formally accepted for publication once it meets all outstanding technical requirements.

Within one week, you’ll receive an e-mail detailing the required amendments. When these have been addressed, you’ll receive a formal acceptance letter, and your manuscript will be scheduled for publication.

Kind regards,

Neftali Eduardo Antonio-Villa, MD PhD

Academic Editor

PLOS ONE

Additional Editor Comments (optional):

I want to congratulate the authors for doing an impressive job in addressing all the suggestions made by the reviewers. The main messages have strength, and I can now recommend this manuscript for publication.
---

## [Editor Report · Acceptance letter]

14 Jan 2025

PONE-D-24-06681R1 

PLOS ONE

Dear Dr. Nonterah, 

I'm pleased to inform you that your manuscript has been deemed suitable for publication in PLOS ONE. Congratulations! Your manuscript is now being handed over to our production team.

Kind regards, 

on behalf of

Dr. Neftali Eduardo Antonio-Villa 

Academic Editor

PLOS ONE